# The Origin of Evergreen Broad-Leaved Forests in East Asia from the Evidence of Floristic Elements

**DOI:** 10.3390/plants13081106

**Published:** 2024-04-16

**Authors:** Hua Zhu, Yunhong Tan

**Affiliations:** Center for Integrative Conservation, Xishuangbanna Tropical Botanical Garden, Chinese Academy of Sciences, Yunnan International Joint Laboratory of Southeast Asia Biodiversity Conservation, Yunnan Key Laboratory for the Conservation of Tropical Rainforests and Asian Elephants, Mengla 666303, China; tyh@xtbg.org.cn

**Keywords:** floristic elements, origin and evolution, subtropical evergreen broad-leaved forest, East Asia

## Abstract

Arguments about the origin and evolution of the evergreen broad-leaved forests in East Asia exist generally, and are even contradictory in some cases. The origin and evolution of the flora of East Asia, especially in the evolutionary process, the formation time of the Asian monsoon, the implications of phylogenetic and biogeographic studies on some important taxa, and the implications of palaeobotanical evidence are debatable. Most research from different disciplines suggests that the monsoon in the Miocene was key to the diversification of East Asian flora and its evergreen broad-leaved forests. The common view is that the evergreen broad-leaved forests of East Asia are closely related to the monsoon’s intensity and developments, which were caused by the uplift of Himalaya–Tibet during or after the mid-Miocene. Analysis of the floristic elements show that the present subtropical evergreen broad-leaved forests in East Asia could have an early or ancient tropical origin and a tropical Asian affinity, but that their species are dominated by endemic Chinese or East Asian ones, many of which have tropical Asian affinity or are tropical sister species. The time of Himalayan uplift and the intensity of the monsoon climate are believed to be key to the formation of the evergreen broad-leaved forests in East Asia. Combined with existing paleobotanical findings, the uplift of the Himalayas and the formation of the monsoon climate, as well as floristic elements of the subtropical evergreen broad-leaved forests, we believe that they evolved from an Asian tropical rainforest after the mid-Miocene in the southeastern region of East Asia, while the ancient subtropical evergreen broad-leaved forests in the southwestern region continuously evolved into the present subtropical ones.

## 1. Introduction

Subtropical evergreen broad-leaved forests in East Asia are distributed generally from the Tropic of Cancer northward to the southern slope of the Qinling Mountains, with a north latitude of 31–32° in China and a north latitude 36° in the southernmost part of Japan and Korea [1]; they are continuously distributed an east longitude of 85° in central and eastern Nepal to Taiwan Province in southeast China [2,3,4,5,6]. Their distribution in China is illustrated in Figure 1. These subtropical evergreen broad-leaved forests are known as “lucidophyllous forests” in Japan (trees with thick, lucid leaves) [7], and are also called “temperate evergreen broad-leaved forests” in Europe [2]. However, the subtropical evergreen broad-leaved forests in East Asia occur in monsoon climates alternating between tropical and subtropical humid summers and dry winters, and have tropical floristic components in the subcanopy and understory, indicating that they are unique in the world, differing from the evergreen broad-leaved forests that developed in a dry temperate summer climate in Europe [2].

The subtropical evergreen broad-leaved forest is the core and representative vegetation type in the subtropical zone of East Asia. Its flora contributes the majority of East Asian flora. East Asian flora is one of the largest centers of species diversity, mainly because of the two major floras: tropical Asian flora and the template flora of Eurasia; in these, different geological and phylogenetic histories meet. East Asian flora was once believed to be a cradle of angiosperms [4,8], but molecular phylogenetic evidence suggests that this cradle is West Gondwana, i.e., South America and West Africa [9]. Exploring the origin of East Asian flora has becomes a point of research interest. The subtropical evergreen broad-leaved forests in East Asia are naturally given extensive attention.

However, arguments about the evolutionary process of East Asian flora and the formation of its subtropical evergreen broad-leaved forests exist generally, and are even contradictory in some cases of multidisciplinary and even single-discipline research. For example, phylogenetic and molecular biogeographic studies of important taxa have different implications regarding the time of the formation of the subtropical evergreen broad-leaved forests in East Asia, with some in the Paleogene period, and some in the Neogene. In paleobotanical findings, fossils from the middle Miocene also show different vegetations: subtropical evergreen broad-leaved forests similar to the current examples occurred in southwest China (Yunnan), while a tropical rain forest similar to the present one in Southeast Asia occurred in southeast China (Fujian). Clarifying these questions and debates about the origin and evolution of the subtropical evergreen broad-leaved forests in East Asia, and finding the gaps, are the aims of this review article.

We reviewed findings closely related to the origin and evolution of East Asian flora and its subtropical evergreen broad-leaved forests from multidisciplinary literature. In this article, we particularly try to understand the occurrence of the evergreen broad-leaved forests of East Asia from geological and climatic histories, phylogenetic and paleobotanical findings, as well as the less known floristic elements. We believe that this review will draw attention to the intriguing and debatable origin and evolution of East Asian flora and its evergreen broad-leaved forests.

In the methodology of this article, literature closely related to the origin and evolution of East Asian flora and its subtropical evergreen broad-leaved forests was reviewed from multidisciplinary areas, including geological and climatic histories, and phylogenetic and paleobotanical findings, and the most important ones were selected. The floristic elements of the subtropical evergreen broad-leaved forests in East Asia were analyzed at the family, generic and species levels. The patterns of seed plant distribution used in the article were quantified at the generic level based on Wu’s and Wu et al.’s [10,11] documentations, and at the family level following Wu et al. [12]. At the species level, the distribution types or geographical elements were confirmed by their distributions recorded in “Flora of China” [13]. The circumscription of families and species nomenclature and classification also followed the “Flora of China” [13].

## 2. Debates in the Origin and Evolution of the Flora in East Asia

Wu [14] analyzed Chinese seed plant flora at the generic level, giving the conclusion that the Chinese flora had a tropical floristic affinity. In this way, the evergreen broad-leaved forest, as a core evergreen forest in East Asia, has a tropical floristic affinity that is beyond question. In the floristic regionalization of the world [15], the area of East Asian subtropical evergreen broad-leaved forest was delineated as being in the Eastern Asiatic floristic region. Later, Wu and Wu raised this floristic region to a kingdom level, the Eastern Asiatic Kingdom, considering its uniqueness, with more than 30 endemic families and an exceptionally large number of endemic genera [16]. The Eastern Asiatic Kingdom was further divided into Sino-Himalayan (with 144 endemic genera) and Sino-Japanese (with 104 endemic genera) subkingdoms [16]. The Eastern Asiatic Kingdom was supposed to be one of the major centers for the evolution of higher seed plants, as the floristic kingdom is especially rich in gymnosperms and primitive angiosperms [16]. However, arguments regarding the origin of the evergreen broad-leaved forests in East Asia and the floristic subdivision of the Eastern Asiatic Kingdom, referring to their floristic origin, have existed for a long time.

Generally, East Asia is considered to be home to high levels of biodiversity and endemism [17]. Based on the findings in paleoendemic taxa, the East Asian region has also been considered to be a floristic museum [16,18]. It has been suggested that direct competition with angiosperms increased the extinction of conifers [19]. In East Asia, the same has happened, with angiosperms increasing with the failure of conifers.

The mountains of southern China are considered to be both “Plant Museums” and “Plant Cradles” [18]. “Refugia” and a main endemism exist in the Chinese southern mountains. It has been further clarified that young endemics occur more in the mountain ranges of the eastern fringe of the Tibetan Plateau (“plant cradles”), but old endemics tend to occur in the mountains of central, south-central, and southeastern China (“plant museums”), which is believed to be related to the different geological history of the mountain ranges [18]. The eastern fringe of the Tibetan Plateau was mostly formed by the uplift of the Himalayas in the late Neogene [20,21], while central and southern China during most of the Tertiary period experienced tectonic stability. This supports Wu and Wu’s idea that the Sino-Japanese floristic region is older than the Sino-Himalayan region in floristic origin in East Asia [16]. However, recent paleobotanical research has revealed that the Qinghai–Tibet Plateau may have risen earlier, possibly in the Paleogene [22,23,24,25], which challenges previous ideas.

Studies on the evolutionary history of the angiosperm flora of China found that 66% of the angiosperm genera in China did not originate until early in the Miocene epoch (23 million years ago (Mya)) [26]. The flora of eastern China bears a syndrome of older divergence (divergence times of 22.04–25.39 Mya), phylogenetic overdispersion (spatial co-occurrence of distant relatives) and higher phylogenetic diversity; meanwhile, in western China, the flora shows more recent divergence (divergence times of 15.29–18.86 Mya), pronounced phylogenetic clustering (co-occurrence of close relatives) and lower phylogenetic diversity [26]. As such, eastern China represents a floristic museum, while western China is an evolutionary cradle, especially for herbaceous genera; however, for woody genera, eastern China serves as both a museum and a cradle [26]. Chen et al. asked a question: is the East Asian flora ancient or not [27]? They synthesized molecular fossil data on seed plants, focusing on the biogeographical origins and historical evolution of East Asian flora. Their results suggested that East Asian flora might be relatively young, with most of its clades originating after the Miocene. East Asia was a refuge for many ancient relict plants, but not their origin area [27]. The former Sino-Himalayan flora was also renamed *Rhododendron* flora, and the Sino-Japanese flora was renamed *Metasequoia* flora, and it was considered that the *Rhododendron* flora and the *Metasequoia* flora both were probably of a similar age [27]. It has been suggested that the formation and development of the Asian monsoon were the main factors driving the evolution of East Asian flora [27].

The current fossil history of five endemic families (Cercidiphyllaceae, Eucommiaceae, Ginkgoaceae, Sargentodoxaceae and Treochodendraceae) and 20 endemic genera was studied [28]. It was found that these endemic plants have three sources: the Arcto-Tertiary area, the boreotropics and East Asia. Eastern flora are believed to have a complex origin, and the modern East Asian plant kingdom is believed to have been formed in the late Pliocene or early Quaternary period [28].

## 3. Debates in the Formation Time of Asian Monsoon

The Asian monsoon is believed to be a key factor in the origin of the evergreen broad-leaved forests in East Asia. However, the timing of the Asian monsoon formation and intensification is debatable.

Based on the simulation of late Oligocene paleogeographic data, it was suggested that the uplift of the northern Qinghai–Tibet Plateau in the Paleogene enhanced the East Asian monsoon climate system and drove the formation of humid and semi-humid vegetation types, dominated by evergreen broad-leaved forests in East Asia [29]. This idea implied that the evergreen broad-leaved forests in East Asia could have been formed in the Paleogene. The uplift of the Himalayan and Tibetan Plateau and its linkage with the evolution of the Asian monsoon was discussed in [30]. It was suggested that the uplift of the southern and central Tibetan Plateau intensified the Indian summer monsoon at 40–35 Mya, and intensified the desertification of inland Asia at 25–20 Mya; meanwhile, the uplift of the northeastern and eastern Tibetan Plateau further intensified the East Asian summer monsoon and East Asian winter monsoon at 15–10 Mya [30], which implies that the evergreen broad-leaved forests in East Asia were formed in the Neogene. Although there is much debate about the timing of the monsoon climate in East Asia, the past mainstream view is that in the late Eocene, about 45–50 Mya ago, the Indian plate and the Eurasian plate collided and integrated, and the Himalayan–Tibetan Plateau did not strongly rise, but experienced a long process of uplift and deplanation at a low altitude (1000–2000 m); in addition, the Himalayas were not so high until the Quaternary period, before 3.4 Mya or 2.5 Mya [20,21]. However, it was recently proposed that the Qinghai–Tibetan Plateau rose earlier based on paleobotanical research [22,23]. It is believed that part of the Himalayan–Tibetan Plateau had already reached 4600 m in 35 Mya [31].

The impacts of major geological events on Chinese flora were reviewed in [32]. It was stated that the main families of the evergreen broad-leaved forests of East Asia, i.e., Fagaceae, Lauraceae, Magnoliaceae, Fabaceae, and Hamamelidaceae, have been present in China since the Paleogene, and that the floristic composition of fossils in southwestern China are similar to the ones of the present evergreen broad-leaved forests. It was also pointed out that the intensity and development of the monsoon are correlated with the height of the Qinghai–Tibetan plateau, which is the main factor involved in the appearance of the evergreen broad-leaved forests of East Asia [32].

It is a common view that the evergreen broad-leaved forests of East Asia are closely related to the intensity and development of the monsoon, which were influenced by the uplift of Himalaya–Tibet. When, then, did Himalaya–Tibet uplift to enough height to create the monsoon? This is back again to the controversial issue of whether Himalaya–Tibet had already uplifted to a height significant enough for the formation of the monsoon before the Miocene [33], whether Himalaya–Tibet slowly uplifted to the present height in the later Pliocene [34], or whether the Himalayas were not so high until the Quaternary period, before 3.4 Mya or 2.5 Mya [20,21].

During the Cretaceous period, China was high in the east and low in the west, and was in a subtropical high-drought zone [35]. The north–south range expanded from a north latitude of 18 degrees to a north latitude of about 38 degrees, and the climate was hot and dry [35]. During that time, there was not a tropical or subtropical evergreen broad-leaved forest in China. After the late Oligocene–early Miocene (about 26–22 Mya), the Himalayas began to rise strongly, and the Yunnan–Guizhou Plateau was formed; in addition, the Asian monsoon significantly increased and the subtropical arid zone of central and eastern China disappeared, being replaced by a warm and humid Asian monsoon climate [35]. A sporopollen in assemblages recorded in the late Eocene strata of the Jianchuan Basin in southwest China was studied, and it was found that the climate was relatively dry and hot, and the vegetation was a tropical to subtropical sparse forest in that period [36]; later, during the period 40.6–37.5 Mya, there was a prominent decrease in xerophilous plants in the fossil assemblages, which implied a climate with increased humidity, and that the vegetation was an evergreen deciduous broad-leaved tree mixed with a coniferous forest under a subtropical–temperate climate [36]. It was also concluded that the humid monsoon climate may have reached the position of the modern monsoon front by the early late Oligocene [36]. Using a general circulation model and geological data, the drivers controlling the evolution of the monsoon system over the past 150 Mya were explored [37]. It was suggested that, apart from a dry period in the middle Cretaceous, a monsoon system had existed in East Asia since at least the Early Cretaceous [37].

Until now, the timing of the monsoon climate in East Asia, which led to the formation of the evergreen broad-leaved forests in East Asia, has been uncertain. Undoubtedly, the uplift of the Himalayas greatly drove the strength of the southwest monsoon [25].

## 4. Phylogenetic and Molecular Biogeographic Implications from Important Taxa

In recent years, much phylogenetic and molecular biogeography research on genera and species with an East Asian distribution has been published, helping to clarify the origin of East Asian flora. Several examples are selected, such as the following:

Species of white pines was studied [38]. It was found that two main clades of subtropical East Asian white pines first diverged in the early Miocene, and by the late Miocene, all species had appeared [38]. It was suggested that the monsoon-driven assembly of evergreen broad-leaved forests might have significantly affected the diversification of subtropical East Asian white pines [38]. It was also indicated that subtropical East Asia is not only a floristic museum, but also a diversification center for gymnosperms [38]. The evolutionary radiation of the genus *Oreocharis* (Gesneriaceae) has diversified extensively throughout East Asia, especially within the Hengduan Mountains. This genus contains 28 species, of which 27 species are in China, with a few in northern Vietnam [39]. The diversification dynamic of *Oreocharis* is believed to be most likely positively associated with temperature-dependent speciation and dependency on the Asian monsoons [39]. The warm and humid climate of the mid-Miocene, together with the East Asian monsoons and global temperature change, may have functioned synchronously as the primary drivers of diversification in *Oreocharis* [39]. The studies both on white pines and the *Oreocharis* suggested that the monsoon in the Miocene was key to the diversification of East Asian flora and its evergreen broad-leaved forests.

The tea family (Theaceae), a characteristic component of the present subtropical evergreen broad-leaved forests, integrating data from other characteristic components of the subtropical forests, including Fagaceae, Lauraceae and Magnoliaceae, was used to discuss the assembly of Asian evergreen broad-leaved forests [40]. Most of the essential elements of the subtropical evergreen broad-leaved forests appeared to have originated around the Oligocene–Miocene (O–M) boundary, but small woody lineages from Theaceae were dated to the late Miocene [40]. The results of this study suggested that two independent intensifications of the East Asian summer monsoon around the O–M boundary and the late Miocene may have facilitated the historical assembly of the subtropical evergreen broad-leaved forests in East Asia.

*Quercus* section *Cyclobalanopsis*, a dominant lineage in East Asian evergreen broad-leaved forests, was studied [41]. The earliest divergences in section *Cyclobalanopsis* correspond to the phased uplift of the Himalayas and the lateral extrusion of Indochina at the transition of the Oligocene and Miocene, while the highest rate of diversification occurred in the late Miocene [41]. It is believed that the dispersal of *Cyclobalanopsis* from the Sino-Himalaya region and the Palaeotropics to the Sino-Japan region in the Miocene was facilitated by the increased intensity of East Asian summer monsoons and by the Middle Miocene Climatic Optimum [41]. From this case, it was suggested that the East Asian evergreen broad-leaved forests began in the Sino-Himalaya region and dispersed into the Sino-Japan region in the Miocene [41]. *Quercus* section *Ilex* is believed to have been widespread along the East Tethys Sea away from the middle Eocene onward, with the species characterizing the sclerophyllous evergreen broad-leaved forests in East Asia as a particular Tethys-affiliated remnant vegetation [42]. Studies on the origins of the species *Quercus* section *Ilex* (holly oaks) revealed that the section *Ilex* originated in East Asia and dispersed to Europe through a warm, humid evergreen forest corridor in Tibet–Himalaya during the Oligocene [43].

The biogeographical diversification of mainland Asian *Dendrobium* (Orchidaceae) suggests that these evergreen broad-leaved forests have been established in mainland Asia at least since the Oligocene [44]. The time at which the evergreen broad-leaved forests were established in East Asia was suggested to be earlier [44] than in most other research.

East Asia–North America disjunct distribution genera are an important floristic element in the subtropical evergreen broad-leaved forests of East Asia. *Tsuga* (hemlock) is such a genus of Pinaceae with a typical intercontinental disjunct distribution in East Asia and Eastern and Western North America. It often occurs in the mountains above subtropical evergreen broad-leaved forests. A phylogenetic analysis revealed that *Tsuga* very likely originated from North America in the late Oligocene and dispersed from America to East Asia via the Bering Land Bridge during the middle Miocene, and that a complex reticulate evolutionary pattern among the East Asian hemlock species has happened [45]. The clade Benthamidia in the genus Cornus is also a floristic element in the subtropical evergreen broad-leaved forests of East Asia. It is an East Asia–North America disjunct distribution taxa [46]. Based on the construction of a molecular phylotree and the reconstruction of ancestral regions of the clade Benthamidia, it was found that the Benthamidia contains two distinct clades: East Asia and North America. It was also found that the ancestors of Benthamidia diverged in the southern part of East Asia, producing the present-day East Asian clade in the middle Oligocene [46]. These two molecular phylogenetic studies indicated that the East Asian flora has affinities with northern America.

Long-term cooling had a disproportionate effect on non-tropical diversification rates, leading to dynamic young communities outside the tropics, while relative stability in tropical climes led to older, slower-evolving but still species-rich communities [47], which suggests that the present subtropical communities in eastern Asia could be young.

Although molecular biogeographical research on different taxa has revealed different occurrence times and dispersion roads for East Asian flora and its evergreen broad-leaved forests, most researchers have suggested that the evergreen broad-leaved forests in East Asia appeared in the Miocene, driven by the East Asian monsoon.

## 5. Palaeobotanical Evidence

Paleobotanical studies offer good clues to the origin and evolution of East Asian flora. The fossil history of seed plant genera that are now endemic to East Asia was reviewed in [48]. It was found that the majority of now “eastern Asian endemic” genera had fossil records from Europe and/or North America, indicating that East Asia served as a late Tertiary or Quaternary refugium for them, although many of these genera may have originated in other parts of the Northern Hemisphere [48]. However, these genera with fossils that are now endemic to East Asia contribute only a small part of the floristic composition of the evergreen broad-leaved forests in East Asia.

Based on Chinese Neogene data, including 71 palaeobotanical sites from the early Miocene to Pliocene, the Neogene vegetation of southern China was reconstructed [49]. It was found that the broad-leaved evergreen component was greater in the more southern areas, with a further rise in the broad-leaved evergreen component in the southern areas, indicating an increasingly warm and moist climate. The western areas were drier than the eastern ones in the Early Miocene, but this variation vanished in the Late Miocene, and the aridification of the western regions occurred again in the Pliocene [49].

It has been suggested that the evergreen broad-leaved forests of southwestern China started in the later Miocene [50,51,52]. However, in the late Oligocene, except the genus *Trigonobalanus*, all genera of Fagaceae had records in China and dominated in the Neogene fossils, suggesting that the evergreen broad-leaved forests of East Asia may have been present since the Neogene [32].

Based on palaeobotanical and molecular phylogeny studies on the psychrophytes in the Hengduan Mountains, these psychrophytes have built up since the early Oligocene [24]. Some present genera and species, such as *Quercus*, *Alnus*, *Betula*, *Carpinus*, *Carya*, *Pterocaryae* etc., appeared in the Luhe basin in southwestern China (central Yunnan) in the early Oligocene (33–32 Mya) [53]. Subtropical forests have been supposed to exist in the central Qinghai–Tibetan Plateau in 47 Mya [54]. The occurrence of fossil fruits of the strictly tropical Southeast Asian rainforest family Dipterocarpaceae in the middle Miocene in Fujian, southeast China [51,55], suggests that southern Fujian had a tropical rainforest vegetation typical of Southeast Asia at that time [56]. However, fossil sites offer occasionally contradictory implications. For example, a present temperate to subtropical deciduous tree *Parrotia* (Hamamelidaceae) was found in a dipterocarp fossil assembled in the middle Miocene in Fujian, which coexisted with plants of Dipterocarpaceae (*Dipterocarpus*, *Hopea*, *Parashorea*, *Shorea*) [57], but the dipterocarp fossils in the ensemble indicated that the fossil ensemble represented a tropical Southeast Asian rainforest. The temperate deciduous tree *Parrotia* found in the tropical fossil ensemble complicates the situation.

The palaeobotanical evidence reveals that southwestern and southeastern China may have had different vegetation in the Miocene. It has been suggested evergreen broad-leaved forests already existed in southwestern China in the Miocene, but it is clear from palaeobotanical evidence that the tropical rainforest vegetation of Southeast Asia existed at that time in southern Fujian. This complicates the origin of East Asian flora, including the evergreen broad-leaved forests.

## 6. The Floristic Elements of the Subtropical Evergreen Broad-Leaved Forest in East Asia

The vegetation characteristics and species composition of the subtropical evergreen broad-leaved forests were studied using eight dynamics plots of evergreen broad-leaved forests in East China [58]. These eight dynamics plots were Tiantong (29°48′ N, 121°47′ E, 304–603 m altitude), Gutianshan (29°15′ N, 118°07′ E, 446.3–714.9 m altitude), Baishanzu (27°45′ N, 119°13′ E, 1470–1593 m altitude), Badagongshan (29°46′ N, 110°25′ E, 1355–1456 m altitude), Heishiding (23°31′ N, 111°52′ E, 435–698 m altitude), Dinghushan (23°10′ N, 112°31′ E, 230–470 m altitude), Fushan (24°45′ N, 121°33′ E, 650–700 m altitude) and Lianhuachi (23°54′ N, 120°52′ E, 667–845 m altitude). They cover the main subtropical evergreen broad-leaved forest types in East Asia. Based on the species list of the eight dynamics plots [58], the floristic elements were analyzed.

At the family level, it was found that the tropical families contributed 52.69% of the total families; of these, the pantropic families make up 37.63%, which is the highest element, while the temperate families contribute 29.04% (Table 1). This indicates that the subtropical evergreen broad-leaved forests could have an early or ancient tropical origin.

At the generic level, the tropical genera contribute 62.41% of the total genera; of these, the tropical Asian genera make up 19.50%, which is the highest ratio, while the temperate genera, including the ones of East Asia and those endemic to China, contribute 36.52% (Table 1). This reveals that the subtropical evergreen broad-leaved forest has a tropical affinity at the genera level, especially affected by tropical Asian elements (floristic elements at the generic level have been recognized to reflect the floristic affinity in biogeography).

However, at the specific level, the tropical species contribute 25.90% of the total species; of these, the tropical Asian species, including the mainland Southeast Asian distribution, make up 23.78%, while species endemic to China or East Asian elements contribute 66.63%, which make up the highest ratio (Table 2). This indicates that the subtropical evergreen broad-leaved forest is dominated by the East Asian floristic element at the species level. The present subtropical evergreen broad-leaved forest has evolved into a vegetation almost endemic to East Asia.

## 7. Origin and Evolution of the Subtropical Evergreen Broad-Leaved Forest in East Asia

Qian and Ricklefs [17] analyzed the temperate-zone genera of East Asia and eastern North America and found that East Asia has twice the species diversity of North America, although both of them share genera in mostly sister pairs, share a common history of adaptation and had an ecological relationship before disjunction. It was proposed that the anomaly in diversity between East Asia and eastern North America was the extreme physiographical heterogeneity of temperate East Asia [17]. Therefore, the origin of East Asian flora, including that of the evergreen broad-leaved forests, should have a more complicated history.

The geological and climatic histories of a region directly affect the formation and evolution of its flora and vegetation [59,60]. The findings of paleobotanical research in East Asia not only provide a basis for exploring the time of Himalayan uplift and monsoon climate formation, but are key to solving the evolutionary history of regional flora and vegetation. The time of Himalayan uplift and the intensity of the monsoon climate were key to the development of evergreen broad-leaved forests in East Asia from our point of view.

It was revealed that the Sino-Himalayan flora developed from lowland biomes predominantly characterized by tropical floristic elements before the collision between the Indian subcontinent and Eurasia during the Early Cenozoic [61]. The present Sino-Himalayan flora is relatively young, and influenced by the uplift of the Himalayas and Hengduan Mountains and the onset and intensification of the Asian monsoon system [61]. This supported our floristic suggestions, which were that the subtropical evergreen broad-leaved forest could have an early or ancient tropical origin, in both the Sino-Japanese region and the Sino-Himalayan region.

Jacques et al. [49] suggested that the western areas were drier than the eastern ones in the Early Miocene; however, this vanished in the Late Miocene, with all areas having a balanced supply of moisture, and the aridification of the western regions occurred again in the Pliocene. The earliest dipterocarp fossils in East Asia, which indicated a tropical rainforest, were found in the Maoming Basin in southern China (21°70′ N, 110°89′ E) in the Late Eocene [62]. In the middle Miocene, dipterocarp fossils were found in southern Fujian (24°12′ N, 117°53′ E) of southeastern China, which also indicated a tropical rainforest [51,55,56]. It has been revealed that the tropical rainforest of Southeast Asia appeared in southern China in the Late Eocene, and later in southeastern China in the middle Miocene. Based on the paleobotanical findings above, we hypothesize that the vegetation in the southeast and southwest of China may have evolved differently from the Miocene to the Pliocene. Evergreen broad-leaved forests in eastern China may have appeared later than those in western China. Evergreen broad-leaved forests in eastern China could have evolved from tropical rain forests after the middle Miocene.

During the hottest period of the Miocene period, the tropical rainforests of Southeast Asia moved north into southeast China, even to southern Japan [63,64]. This suggests that the modern subtropical evergreen broad-leaved forest developed in southeastern China, with the tropical rainforest retreating to the south after the middle Miocene. On the other hand, in SW China, such as in Yunnan, Guizhou, Sichuan, and even Xizang (Tibet), there are no fossil records of the typical tropical rainforest components (for example, dipterocarp fossils) during the Neocene, but a lot of fossils of the subtropical evergreen broad-leaved forest components have been found. This could match Jacques et al.’s conclusion that the more western Chinese areas were drier than the eastern ones in the Early Miocene, and that the aridification of the western regions occurred in the Pliocene [49]. We believe that the typical tropical rainforest could not have appeared in SW China during the Miocene to the late Pliocene. We suppose that the tropical rainforest vegetation occurred late in Yunnan in southwestern China, no earlier than 3–5 Mya [59,65,66].

Southern Fujian in southeastern China has a typical subtropical evergreen broad-leaved forest, but the tropical rain forest vegetation of Southeast Asia existed in the middle Miocene [51,55,56]; meanwhile, at almost the same time, the palaeobotanical information in southwest China shows that it had a subtropical evergreen broad-leaved forest in the Miocene, similar to the present one [50], meaning that southwestern and southeastern China should have had different climatic and vegetation, at least in the middle Miocene.

Many tropical Asian taxa are more widespread in subtropical areas of East Asia, and some are represented in the subtropical evergreen broad-leaved forest, or are locally endemic species with tropical affinity [2]; this includes, for example, species belonging to tropical or tropical Asian genera such as *Schima*, *Altingia*, *Exbucklandia*, *Rhodoleia*, *Nyssa*, *Lithocarpus*, *Castanopsis*, *Sloanea*, *Symplocos*, *Daphniphyllum*, *Meliosma*, *Illicium*, *Bischofia* and *Adinandra.* These woody species have tropical affinity, but have evolved into subtropical and temperate species in East Asia. They are considered to be tropical sister taxa. As the understory of the subtropical evergreen broad-leaved forests in East Asia, the tropical herbaceous families Gesneriaceae, Begoniaceae, and *Elatostema* of Urticaceae have also evolved a lot of East Asian or Chinese species [2,67].

An example is the so-called tropical rainforest in Jinggangshan, Jiangxi Province (at 26.57° N) [68], which is in the zone of a typical subtropical evergreen broad-leaved forest [69]. From an investigation of the forest in Jinggangshan [68], it has characteristics such as big woody lianas and big trees with buttresses, which are similar to a tropical rain forest. The tree species with high phytosociological importance are *Distylium myricoides*, *Castanopsis lamontii*, *Exbucklandia tonkinensis*, and *Alniphyllum fortune*, which are mostly endemic to China or East Asia, but they are tropical sister species because their genera are mainly have a tropical Southeast Asian distribution.

*Lasianthus* is a large genus of Rubiaceae, predominantly distributing in the Old World tropics. The greatest species diversity was found in tropical Asia [70]. The species of the genus occur almost exclusively in the understory of primary forests, especially tropical rainforests. In this genus, only one species, i.e., *Lasianthus japonicas*, has a typical East Asian distribution. It includes two subspecies: Subsp. *japonicus* occurs in southeastern China to Japan below an altitude of 1000 m, and in the Sino-Japanese floristic region, as delineated by Wu and Wu [16]. Subsp. longicaudus occurs on montane areas over an altitude of 1000 m in southwestern China to the Himalayan region, i.e., the Sino-Himalayan floristic region [13,71,72]. The formation of the vicarious distribution patterns of the Sino-Himalayan and Sino-Japanese regions is supposed to be related to the uplift of the Himalayas in the Tertiary period. *Lasianthus japonicus* subsp. longicaudus could have differentiated as an altitudinal vicariant of L. japonicas with the uplift of the Himalayas. As an understory shrub of the tropical rainforest, the genus *Lasianthus* and many other shrub and herbaceous plants of tropical genera, although they are presently endemic East Asian species and exist commonly in the understory of subtropical evergreen broad-leaved forests, they are the sister species of the ones in the tropical rainforest flora of Southeast Asia. These tropical sister species have evolved, driven by the geological history of East Asia.

*Magnolia* section *Michelia* (Magnoliaceae) is distributed in tropical southeast Asia to East Asia. The taxa of *Michelia* have been dated to the late Oligocene in southeast Asia, but the diversification of its core group (taxa) occurred mainly from the late Miocene onward in East Asia [73]. The result of the genus *Michelia* is similar to *Dendrobium* (Orchidaceae) [44]. Our floristic evidence suggests that the subtropical evergreen broad-leaved forest, especially in the subtropical eastern part of East Asia, derived from a tropical southeastern Asian flora, and developed in the late Miocene onward.

From the analysis of the floristic elements of the subtropical evergreen broad-leaved forest above, especially in the Sino-Japanese region of East Asia, it is obvious that the present subtropical evergreen broad-leaved forests could have an early or ancient tropical origin at the family level, and have a tropical Asian affinity at the generic level. The subtropical evergreen broad-leaved forest is dominated by endemic species with a Chinese or East Asian distribution, as well as tropical sister ones (with tropical affinity). A large number of tropical shrubs and herbaceous plants are present in the understory of the subtropical evergreen broad-leaved forests in the Sino-Japanese region of East Asia, and a lot of the tropical sister tree species are present in the sub-canopy of the forests [2,67], which demonstrate the tropical affinity of the subtropical evergreen broad-leaved forests in East Asia.

Combined with palaeobotanical findings, the uplift of the Himalayas and the formation of a monsoon climate, we believe that the subtropical evergreen broad-leaved forests in the Sino-Japanese region of East Asia evolved from a tropical Asian rainforest after the mid-Miocene.

## 8. Discussion

The East Asian flora and its evergreen broad-leaved forest are almost consistently believed to be strongly affected by geological history and the formation of the monsoon climate, especially the event of the uplift of the Himalayas and the consequent monsoon formation. Arguments focus on the timing of the uplift of the Himalayas and the consequent monsoon formation, which were during the Paleocene or the Neocene. Different theories on geological histories and the formation of the monsoon climate, incomplete palaeobotanical findings, the evolutionary history of the angiosperm flora of China, and research on the phylogenetic biogeography of important taxa of East Asian distribution mean that suggestions and implications regarding the origin and evolution of East Asian flora and its evergreen broad-leaved forests are not consistent. The East Asian flora was even suggested as a refuge of many ancient relict plants, but not their origin area. Regardless of the debates, most research from different disciplines has suggested that the monsoon climate in the Miocene was key to the diversification of East Asian flora and the origin of its evergreen broad-leaved forests.

A key lineage in the subtropical evergreen broad-leaved forest in East Asia, *Litsea* complex (Lauraceae), was phylogenomically analyzed, combining the geological and paleontological records and the Cenozoic climate change [74]. It was revealed that the prevailing of the East Asian monsoon system in the Early Miocene accelerated the evolution of evergreen habits and ultimately contributed to the modernization of the subtropical evergreen broad-leaved forest in East Asia [74]. Although we approve of this suggestion, the palaeobotanical evidence reveals that southwestern and southeastern China had different vegetation in the Miocene.

The subtropical forests similar to the present ones in East Asia were suggested to have existed in southwestern China, especially in Yunnan, since the Oligocene [50,51,52,53,54]. The occurrence of fossil fruits of the strictly tropical Southeast Asian rainforest family Dipterocarpaceae in the middle Miocene in Fujian, southwestern China [55,56], suggested that a tropical rainforest of Southeast Asia existed there at that time. The palaeobotanical evidence revealed that southwestern China could have had a subtropical evergreen broad-leaved forest in the Miocene, but a tropical rainforest of Southeast Asia in southeastern China at that time. We believe that southwestern China and southeastern China may have had different vegetation in the middle Miocene: a subtropical evergreen broad-leaved forest existed in southwestern China, but a tropical lowland rainforest existed in southeastern China.

From geological history, we accept Guo et al.’s suggestion that during the Cretaceous period, China was high in the east and low in the west, and was in a subtropical high-drought zone [35]. There was not a tropical or subtropical evergreen broad-leaved forest at that time. Since the late Oligocene–early Miocene (about 26–22 Mya), the Himalayas began to rise strongly, and the Asian monsoon climate significantly increased; in addition, the subtropical arid zone of central and eastern China disappeared, replaced by a warm and humid Asian monsoon climate [35]. Some floristic elements of the subtropical evergreen broad-leaved forests appeared in the Oligocene [75], and subtropical evergreen broad-leaved forests similar to the present ones formed during the Miocene in southwestern China [76,77], but a tropical rainforest similar to the ones in present Southeast Asia appeared in south and southeastern China during the hottest period of the Miocene period [59]. After the Miocene, the modern subtropical evergreen broad-leaved forests could evolve as the tropical rainforest retreated to the south margin of southeastern China; meanwhile, the ancient subtropical evergreen broad-leaved forests continuously evolved into the present ones in southwestern China. We also suggest that the typical tropical rainforest could not have appeared in southwestern China from the Miocene to the late Pliocene. We suppose that the tropical rainforest in Yunnan occurred later, no earlier than 3–5 Mya [59].

The analysis of the floristic elements of the subtropical evergreen broad-leaved forest in the Sino-Japanese region of East Asia shows that it has a tropical affinity at the genera level, especially tropical Asian affinity. The present subtropical evergreen broad-leaved forest in this region is dominated by endemic East Asian and Chinese species. Many species, including almost all life forms in the subtropical evergreen broad-leaved forest, are endemic to China but tropical Asian sister species. We could infer that the subtropical evergreen broad-leaved forest in the Sino-Japanese region evolved from a tropical Asian rainforest after the mid-Miocene.

## 9. Conclusions

The origin and evolution of the East Asian flora and its subtropical evergreen broad-leaved forests have been debated in multidisciplinary and even in single-discipline research. Although the timing of the uplift of the Himalayas driving the formation and intensification of the Asian monsoon is still uncertain, the Asian monsoon is believed to be a key factor in the formation of evergreen broad-leaved forests in East Asia. Based on the finding of fossil species of the strictly tropical Southeast Asian rainforest in the middle Miocene in southeastern China, we believe that the present subtropical evergreen broad-leaved forest in the Sino-Japanese region of East Asia appeared after a tropical rainforest, while the fossils of the components of subtropical evergreen broad-leaved forests appearing in southwestern China (Yunnan) since the Oligocene suggest that the ancient subtropical evergreen broad-leaved forests in southwestern China continuously evolved into the present ones.

The floristic elements of the present subtropical evergreen broad-leaved forest in the Sino-Japanese region reveal that it has a tropical Asia affinity, and many of its species, although endemic to China and East Asia, are tropical sister species. This supports that the subtropical evergreen broad-leaved forest in the Sino-Japanese region of East Asia evolved from a tropical Asian rainforest after the mid-Miocene.

## Figures and Tables

**Figure 1 plants-13-01106-f001:**
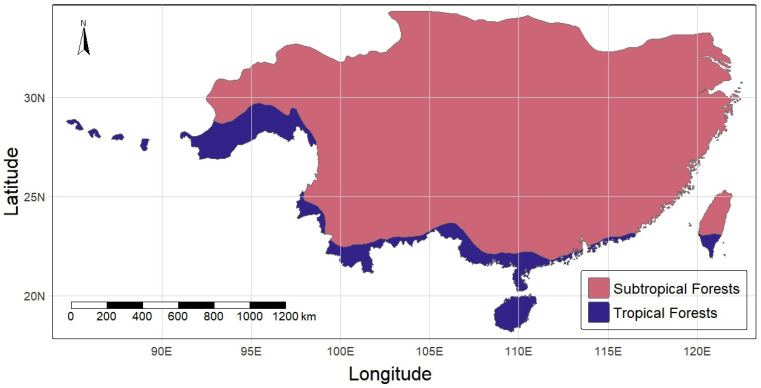
Distribution map of the subtropical evergreen broad-leaved forests in China. The map was drawn by Jian Zhang from the School of Ecological and Environmental Sciences, East China Normal University. Notes: The red patches show the distribution of subtropical forests in China, of which the subtropical evergreen broad-leaved forest, as the zonal forest vegetation type, occurs in lowland areas below 1000 m above sea level, while coniferous forests and coniferous–evergreen broad-leaved mixed or coniferous–deciduous broad-leaved mixed forests occur on mountains above 1000 m above sea level. The blue patches show the distribution of tropical forests, of which the tropical rain forest and tropical monsoon forest, as the zonal forests, occur in lowland areas below 900 m above sea level, while tropical montane evergreen broad-leaved forests and tropical coniferous forests occur on mountains above 900 m above sea level.

**Table 1 plants-13-01106-t001:** Biogeographic elements of seed plant taxa at the family and generic levels of the subtropical evergreen broad-leaved forest (eight permanent plots) in East Asia.

Biogeographical Elements	No. of Families	Family % *	No. of Genera	Genera (%) *
Cosmopolitan	17	18.28	3	1.06
Pantropic	35	37.63	50	17.73
Tropical Asia–Tropical America disjunct	9	9.68	15	5.32
Old World tropic	1	1.08	19	6.74
Tropical Asia to Tropical Australia	2	2.15	28	9.93
Tropical Asia to Tropical Africa	0	0	9	3.19
Tropical Asia	2	2.15	55	19.50
(Tropical in all)	(49)	(52.69)	(176)	(62.41)
North temperate	16	17.2	32	11.35
East Asia–North America disjunct	5	5.38	25	8.87
Old World temperate	0	0	4	1.42
Temperate Asia	0	0	1	0.35
Mediterranean, Western Asia to Central Asia	0	0	1	0.35
East Asia	5	5.38	31	10.99
15 Endemic to China	1	1.08	9	3.19
(Temperate in all)	(27)	(29.04)	(103)	(36.52)
Total	93	100	282	100.00

* Percentage was calculated by the number of families/genera in each geographical element divided by the number of genera of all geographical elements, then multiplied by 100%.

**Table 2 plants-13-01106-t002:** The distributional patterns of seed plant species in the eight permanent plots from subtropical evergreen broad-leaved forests in East Asia.

Distributional Patterns at the Specific Level	No. of Species	Species (%) *
I. Old World Tropic	4	0.50
II. Tropical Asia to Tropical Australia	13	1.62
III. Tropical Asia (India-Malesia) to China or E Asia	(89)	(11.08)
IIIa. to China	62	7.72
IIIb. To East Asia	27	3.36
IV. Mainland SE Asia to China or to E Asia	(102)	(12.70)
IVa. Mainland SE Asia to China	66	8.22
IVb. Mainland SE Asia to East Asia	36	4.48
V. S Himalayas via Mainland SE Asia to China or E Asia	(58)	(7.22)
Va. S Himalayas via Mainland SE Asia to China or E Asia	46	5.73
Vb. S Himalayas to China or E Asia	12	1.49
VI. Endemic to China or East Asia	(535)	(66.63)
VIa. Endemic to China	410	51.06
VIb. Endemic to East Asia (Northeast to Japan and or Korea)	125	15.57
VII. SW Asia, the Mediterranean to China	2	0.25
Total	803	100.00

* Percentage was calculated by the number of species in each geographical element divided by the number of species of all geographical elements, then multiplied by 100%.

## Data Availability

The floristic analysis in this article is from the data on the species list of the eight dynamics plots published by Song et al., 2015 [58]. http://www.biodiversity-science.net/fileup/PDF/w2014-140-1.pdf, accessed on 14 April 2024.

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
