# Peer review of "The Origin of Evergreen Broad-Leaved Forests in East Asia from the Evidence of Floristic Elements"

_plants, 2024, doi:10.3390/plants13081106_

Round 1

Reviewer 1 Report

Comments and Suggestions for Authors

The authors `present an interesting theoretical manuscript dealing with the origin of evergreen broad-leaved forests in eastern Asia. Structured in seven blocks, the authors discuss about the debates on the origin and evolution of the East Asia flora, formation time of Asian moonsoon and paleobotanical evidences among others.

The structure of the ms is well done, and every of the seen blocks is well balanced. The discussion and conclusion block close perfectly the esence of the research. 

In order to imporve the ms I only would suggest to convert the tables 1 and 2 in a single table.ñ

Congratulations for the research.

Author Response

Thank reviewer’s constructive suggestions.

We have combined the table 1 and table 2 into one table in the revised version.

Reviewer 2 Report

Comments and Suggestions for Authors

the article is a careful and interesting review which brings together the various knowledge from the point of view of floristic, geological, climatic, paleobotanical, etc. to make some interesting conjectures about the origin of the Eastern Asia evergreen forest. It is well written and discussed. My suggestion, to make it clearer and more appealing to readers who are not familiar with the territories examined, is to include at least one figure that highlights the geographical areas in relation to the kingdoms and floristic areas mentioned, the direction of the monsoons etc.

Some typos in the attached file

Author Response

Thank reviewer’s constructive suggestions.

In this revision of the article, we added an introduction section, in which the distribution of the subtropical evergreen broad-leaved forests in East Asia were clarified in text and illustrated in added Figure 1.

Typing errors in text were corrected.

Reviewer 3 Report

Comments and Suggestions for Authors

Dear Authors,

The manuscript discusses the research on the origin and evolution of Asian evergreen broad-leaved forests, presenting results from paleobotanical and molecular biogeographical studies, as well as analysis of the characteristic flora of evergreen broad-leaved forests, including their role in the subtropical zone. Overall, the analyses suggest a close relationship between the origin and evolution of evergreen broad-leaved forests and the origin and history of Asian flora, particularly regarding the uplift of the Himalayas and the formation of the Asian monsoon. However, the manuscript lacks a comprehensive summary connecting the results of paleobotanical, molecular biogeographical, and geological studies, which would provide a clearer picture of the origin and evolution of evergreen broad-leaved forests. Further critical analysis is needed to better understand the complex evolutionary processes of these forests and to identify potential gaps or contradictions in current knowledge.

The manuscript is rich in content and includes many interesting results and analyses from research on the origin and evolution of Asian evergreen broad-leaved forests. However, there are some critical observations from the perspective of general scientific communication. The presentation could be shortened and structured to make it easier for readers to understand the results. A general introduction to the topic would be very beneficial. Currently, it feels like the manuscript jumps straight into the main points, potentially causing an average reader to lose interest in the importance and relevance of the topic. Some paragraphs in the introduction are repeated multiple times. Additionally, it would be better if the introduction briefly but clearly summarized the background, objectives, and significance of the research. The discussion section contains many analyses, but lacks a summary section connecting the results and the main goal of the research, as well as the literature context. It would be better if the discussion section included more critical analysis and summarization of the results interpretation and research significance.

One thing that I miss greatly from the manuscript is maps. In my opinion, using maps could not only make the manuscript more visually appealing, but also significantly facilitate explanation and understanding. It might be worth considering using maps for each subsection, providing good visualization. Maps showing spatial and temporal changes could also be necessary. Moreover, it might be useful to make these maps available in SHP file format, facilitating potential further analyses. I strongly recommend supplementing the manuscript with these additions. A potential approach for this could be inspired by the following article: Morrone, J. J., & Márquez, J. (2001). Halffter’s Mexican Transition Zone, beetle generalized tracks, and geographical homology. Journal of biogeography, 28(5), 635-650.

Furthermore, including figures showing the main types of vegetation, typical endemic or relict species could also enhance the manuscript.

The self-citation rate is close to 17%, which I still consider acceptable.

Minor corrections:

Line 36, 78, 98, 105, 170, 195: double space.

L65, 73, 145, 148, 159, 197, 272, 359, 466: a space is missing.

If the authors are named in the manuscript, the reference number must be placed directly after the authors' names. Check it in the full manuscript. For example: Wu and Wu [3], Qian and Ricklefs [4], or Jacques et al. [36].

L51: For example the “Plant Museums” and “Plant Cradles” can be depicted in a map.

L52-56: Keep order: explain the Plant Museum first, then the Plant Cradle.

L54 and L55: The term Plant Museum and Plant Crandles are used consistently in the MS (capitalized if it started this way).

L59: Wu and Wu, instead of Wu & Wu

L157: what do you mean by white pines? Is it a vegetation type or species?

L162: Plant Museum is equal to floristic museum? This should be clarified before because it is confusing.

L179: whose results? The authors should be named at least once “Their results suggested..”

L244: Please Italicized: Trigonobalanus

L311: Qian and Ricklefs, instead of Qian & Ricklefs

L373: Begoniaceae does not Italicized

Overall, the manuscript contains appropriate and valuable scientific content, and would be acceptable with the specified modifications. These errors could easily be rectified through minor corrections and text editing. Adding some additional references or details could also be advantageous to enhance the research context and completeness.

Comments on the Quality of English Language

The language used in the manuscript can generally be considered appropriate, as it is clear and understandable. However, there are some instances of wording and structural errors that need to be addressed to ensure the texts are scientifically accurate and readable. For example, sentences are sometimes long and convoluted, which can hinder readability and comprehension. It would be better to break these sentences into smaller ones or rephrase them to make them shorter. Additionally, the structure and transition of some sentences are not clear, which could be confusing for readers. Furthermore, some expressions and word choices are not always appropriate, and enriching the text with idiomatic expressions or synonyms could be beneficial.

Author Response

Thank reviewer’s constructive suggestions.

In revise version of this review article, we added an introduction section, in which we gave a concise summary to the search results from related literatures, to clarify these existed questions and debates in the origin and evolution of the subtropical evergreen broad-leaved forest in eastern Asia.  These are in paragraphs 3 and 4 of Introduction.

We have shortened the text and deleted repeated sentences as possible as we could. In some places, we rewrite the sentences to try to explain our meaning more clearly.

We split the discussions and conclusion section into discussion section and conclusion section in the revision, so that the discussion section focuses more discussion issues, while the conclusion section summarizes the finding results.

In the beginning of this article, in introduction section, we added a map, i.e., Figure 1, which illustrates the distribution of the subtropical evergreen broad-leaved forest in China. Here I would like to say that Chinese administration has regulations on map use in articles. The map in the review article is the one that we could offer, and more details in the map have to omit on the regulations. We are sorry for this.

Other minor revisions by reviewer’s suggestions have been done in this revision version.

English polishing will be done through professional agency after the article is accepted.

Reviewer 4 Report

Comments and Suggestions for Authors

The peer-reviewed article is in line with the scope of the journal and addresses scientifically relevant questions. Despite all the strengths of the article, it has significant weaknesses. 

1. Despite being a review article, it should start with an introduction that sets the issue in a broader geographical and historical context. The aims and objectives of the review should also be clearly stated, which is essential to enable the rest of the article to follow a clear logical sequence. 

2. I very much miss the methodology section. Sometimes review articles may not have a methodology section, but in this article I think it is essential. The methodology section should state what criteria were used for the selection of the literature and what period the sources covered. It is very important to indicate what generic nomenclature has been followed: whether the generic names used in the original sources have been altered in any way, adjusted to the current nomenclature, or left as they were. All these questions are important for the reader and their explanation is necessary.

3. I very much missed the illustrative material in the biogeographical article, especially the maps. 

4. The text needs significant editing. Many sentences are awkward or even confusing. For example, in lines 366-370, the term 'tropical' is repeated 5 times, making it unclear what the authors intended to emphasise in this sentence. 

The restructured and significantly expanded article needs to be re-evaluated in detail. 

Comments on the Quality of English Language

Moderate editing is required.

Author Response

Thank reviewer’s constructive suggestions.

In revise version of this review article, we added an introduction section, in which we gave a concise summary to the search results from related literatures, to clarify these existed questions and debates in the origin and evolution of the subtropical evergreen broad-leaved forest in eastern Asia. The aims and objectives were included in the introduction section.

About the methodology, we included also in the introduction section as the last paragraph, considering the article is a review, so we did not set a method section.

In the beginning of this article, in introduction section, we added a map, i.e., Figure 1, which illustrates the distribution of the subtropical evergreen broad-leaved forest in China. Here I would like to say that Chinese administration has regulations on map use in articles. The map in the review article is the one that we could offer, and more details in the map have to omit on the regulations. We are sorry for this.

We rewrite some sentences mentioned by reviewer to try to explain our meaning more clearly.

English polishing will be done through professional agency after the article is accepted.

Round 2

Reviewer 3 Report

Comments and Suggestions for Authors

The Authors have properly corrected the manuscript, so I recommend it for publication.

a

rövidítés: működÅ‘, ható, aktív, A-hang
fÅ‘név: jeles osztályzat

Author Response

We did minor revision in the second revision version.

For the English polishing to the article, because it is a charging service, our institute has a requirement that we could do after the article is accepted.

Reviewer 4 Report

Comments and Suggestions for Authors

Revised after reviewers' comments, most of the comments have been taken into account. It is regrettable that the paper focuses on the evergreen broadleaf forests of East Asia and that the map (Figure 1) only covers China, i.e. only part of the territory where the forests in question are located. Why are the other countries in the region and the boundaries of the biogeographical divisions covered in the article not shown on the map? It is very difficult to believe the authors' assertion that there is a restriction on the publication of maps at this scale in scientific publications. What is the basis for the restriction if detailed maps of the entire region can be accessed on the Internet? After all, the recommended map would be very generalised and would not reveal any state secrets. A map showing the biogeographical divisions mentioned in Table 2 would be very important for the article. This is my main point, which is why I am not convinced by the authors' reply. I therefore suggest significant corrections, but the final decision on my recommendation should be taken by the editor. 

The English in the newly drafted introduction of the article needs quite significant editing. The stylistic problems start with the first sentence of the introduction (line 29). 

Comments on the Quality of English Language

Moderate editing is required. 

Author Response

Thank reviewer’s very kind suggestions.

Recent years, we received inform from our administrative office about map use in scientific articles. The inform said that if the map refers to Country borders, the application for the map use in articles will be send to the National Map Secrecy Committee for approve before use. This is a time consuming thing, and difficult to be approved. We guess this situation could be the recent political intensity in eastern Asian countries. We must observe the policy. Therefore, we could offer only simple sketch map. We are regretful and sorry for this.

In Figure 1, we further explain the distribution of vegetation types in these showed areas, as in the notices (labeled by red words in the text in this revising version).

For the English polishing to the article, because it is a charging service, our institute has a requirement that we could do after the article is accepted.

Round 3

Reviewer 4 Report

Comments and Suggestions for Authors

I understand and accept the explanation given by the authors regarding the use of maps in the articles, and it is regrettable that such obstacles have been created to the dissemination of scientific information. 

I have not found any significant deficiencies in the revised version of the article, apart from a few technical inaccuracies, which I believe will be addressed in the final version of the article.

Comments on the Quality of English Language

Minor editing is required.

Author Response

Thank reviewer’s careful and kind comments.

The authors of this article are Chinese natives and are not good in English writing. We suppose that the technical inaccuracies in the manuscript should be in the aspects of our English expressions. We, ourselves could not improve the article in English writing at current stage even we want. This is our weakness. Because English polishing is a charging service, we could not easy to do this before the article is accepted. We are warrant that the English polishing to the article will be sent to do as soon as the article is accepted by the Journal. We hope that the technical inaccuracies in the article could be resolved during English polishing.

     In this version, we checked the cited literatures in the text and in the reference section, keeping their consistency around. We also replaced two references by more closely related to the text (marked in red in reference list). Minor typing mistakes in the article were corrected.
